# The Importance of Rhythmic Stimulation for Preterm Infants in the NICU

**DOI:** 10.3390/children8080660

**Published:** 2021-07-29

**Authors:** Joëlle Provasi, Loreline Blanc, Isabelle Carchon

**Affiliations:** 1Cognitions Humaine et Artificielle -EPHE-PSL, CHArt Laboratory, 93322 Aubervilliers, France; isabelle.carchon@ephe.psl.eu; 2Psychomotor Therapist, DE ISRP, 92100 Boulogne-Billancourt, France; blanc.loreline@gmail.com

**Keywords:** rhythm, preterm infants, sensorimotor synchronization, intrauterine, rhythmical stimulation, infant-directed singing

## Abstract

The fetal environment provides the fetus with multiple potential sources of rhythmic stimulation that are not present in the NICU. Maternal breathing, heartbeats, walking, dancing, running, speaking, singing, etc., all bathe the fetus in an environment of varied rhythmic stimuli: vestibular, somatosensory, tactile, and auditory. In contrast, the NICU environment does not offer the same proportion of rhythmic stimulation. After analyzing the lack of rhythmic stimulation in the NICU, this review highlights the different proposals for vestibular and/or auditory rhythmic stimulation offered to preterm infants alone and with their parents. The focus is on the beneficial effects of auditory and vestibular stimulation involving both partners of the mother–infant dyad. A preliminary study on the influence of a skin-to-skin lullaby on the stability of maternal behavior and on the tonic emotional manifestations of the preterm infant is presented as an example. The review concludes with the importance of introducing rhythmic stimulations in the NICU.

## 1. Introduction

Sensory motor synchronization is crucial for one’s daily cognitive and social activities: walking, writing, dancing, creating music, etc. Furthermore, it seems essential for language and communication skills [1]. Sensory motor synchronization is also crucial for social interaction because it is an efficient way to communicate that one has received the signals from another person’s behavior [2]. The ability to produce an action that intentionally synchronizes with another person’s action is typically learned at a young age. Rhythmic synchrony becomes just one piece in a whole that combines emotion, sociality, and rhythm [3]. In this paper, we highlight the importance of the different rhythms present from the beginning of fetal life and examine the continuity or discontinuity of the various rhythms in the intrauterine environment compared with the incubator in the neonatal intensive care unit (NICU). We illustrate possible solutions to promote sensory–motor synchrony in the preterm infant and thus stimulate interactions. For this purpose, we first present an extensive review of early rhythmic experiences and then the results of a pilot study comparing mother and infant behaviors in kangaroo care with and without maternal singing, i.e., considering the effects of a condition potentially affording additional synchronization cues.

Sensory stimuli are essential for the development of sensory modalities. This explains why some modalities develop earlier than others, depending on the nature and frequency of the stimuli present in the environment. Sensory stimuli play an important role in the initiation, consolidation, modulation, construction specificity, and functionality of neural connections. Thus, environmental stimuli orient the neuro-cognitive development of the unborn child [4]. The fetus can definitively perceive sounds and movements during the third trimester of gestation, i.e., the last three months before birth, due to which there is an increase in fetal cortical brain activity in response to species-typical sounds [5]. The fetus lives in an environment where some rhythms, such as maternal heartbeats and maternal breathing, are omnipresent, without discontinuity, while others, such as the mother’s voice, words, songs, music, and footfalls, are not emitted continuously, without interruption, but occur each time the mother speaks, sings, or walks. 

### 1.1. Rhythm Perception In Utero

#### 1.1.1. Maternal Heartbeat

The various sounds in the uterus are, for the most part, of low frequency and amplitude. However, these low-frequency maternal sounds, such as heartbeats, are audible to the fetus in utero from the beginning of gestation [6]. The maternal heartbeat is audible to the fetus because it is 25 dB higher than the background noise and thus dominates the fetal environment [7], as confirmed by new techniques, such as magnetoencephalography [8]. The fetal heart rate changes based on the mother’s level of activity and stress [9]. The fetal heart rate significantly increases when the mother with above-average anxiety performs a stressful task for 5 min, whereas the fetal heart rate does not significantly change when the mother with below-average anxiety performs the same stressful task. The maternal heartbeat creates a pressure wave that can be not only clearly heard but also felt by the fetus [10]. The continuous, rhythmic sound of the maternal heartbeat is the most prominent and frequently heard stimulus in utero [11]. The maternal heartbeat is the fetus’s first metronome and can influence subsequent preferences for other periodic auditory stimuli [11]. The maternal heartbeat is the first regular and periodic stimulus the fetus receives, so it can influence subsequent preferences generalized to many other slow rhythmic sounds [12]. The same short-term calming effects can be elicited in newborns by other rhythmic noises (e.g., resting heartbeat, lullabies, and even metronome clicks at 72 beats per minute [13]) with acoustic characteristics similar to the maternal heartbeat [14].

#### 1.1.2. Maternal Breathing

Few studies have explored the fetal perception of the maternal respiratory rhythm. However, intrauterine recordings of humans and animals have confirmed that prenatally audible sounds include rhythmical breathing [15]. To demonstrate maternal–fetal heart rate synchronization, Van Leeuwen et al. [16] proved that the fetus perceives the maternal breathing rate and is sensitive to change in its rhythm. The authors observed maternal–fetal heart rate synchronization when the mother breathed at a spontaneous rate of 12 bpm. This synchronization is much more important when the mother’s breathing becomes faster (20 bpm). The fetus can modify its own heart rate to synchronize with the mother’s heart rate, depending on the mother’s respiratory rate. During the last trimester of pregnancy, the fetus perceives changes in the mother’s breathing rhythms and reacts by synchronizing its heart rate to that of its mother [16].

#### 1.1.3. The Mother’s Voice

Sounds in the pregnant uterus vary, with low frequencies and amplitudes that are nonetheless detectable by the fetus [17]. The fetus is most likely exposed to a much richer repertoire of external sounds than was previously thought. When the mother speaks or sings, the fetus can hear the mother’s voice from both an internal and an external source, as the sound originates internally but is also transmitted via air [14]. Despite some loss in the tonal quality of higher frequencies, the mother’s voice is easily detected by the fetus due to its consistent prosody (the melodic contours, accents, and rhythms of language) [12]. The rhythmic properties of language play a major role in the development of language discrimination. Rhythm processing has further been shown to be especially important for language processing and recognition [18]. At an early stage of development, infants perceive speech sounds as music and are likely to attend to the melodic and rhythmic aspects of speech [19]. Tempo changes are already detectable by the fetus [20]. Fetuses were able to discriminate between changes in musical tempo, as evidenced by their behavioral and physiological responses [21]. A 32-week-old fetus used these prosodic cues to differentiate its mother’s voice from another woman’s voice, both of which were broadcast over a loudspeaker [22], and also to differentiate its native language from a foreign language [23]. At birth, newborns are able to differentiate their native language from a foreign language, but they can also differentiate two foreign languages if the languages belong to different rhythmic classes [24]. Electroencephalogram studies reveal that newborns detect rhythm and tempo and the violation of these temporal patterns [25]. Rhythm and tempo allow the listener to break down the flow of language into a meaningful discourse.

#### 1.1.4. Maternal Walk

The rate of 100 steps/min represents a reasonable floor value indicative of moderate-intensity walking in adults [26]. In a review of 32 studies published between 1980 and 2000, Tudor-Locke and Myers [26] indicated that healthy younger adults (approximately 20–50 years old) take 7000–13,000 steps/day. Although pregnancy leads to a decrease in physical activity [27], pregnant women walked an average of 5259 (SD = 1762) steps/day [28], which underlines that walking, a rhythmic activity, remains present during pregnancy. The Canadian guideline for physical activity throughout pregnancy [29] recommends physical and sports activity during pregnancy for obstetrical, general, and psychological benefits. A few studies show that this practice has long-term consequences for children, including improved psychomotor development at 2 years of age [30]. The sound of maternal footsteps has a soothing effect on newborns, demonstrated in classic rocking situations [31]. The fetus reacts by variations of its own cardiac rhythm in response to the multiple sources of rhythmic and vestibular stimulation from the mother and particularly her locomotor movements [32]. When the mother walks, her body movements create numerous rhythmic episodes of angular accelerations that can be detected by the fetus and elicit fetal responses. Variations in the fetal heart rate have been observed in response to the mother’s passive movement: the fetus perceives the difference (evidenced by a significant fetal heart rate deceleration) between the rocking when the mother is in a rocking chair versus a swing [33,34]. The cardiac reactions of the fetus close to term show that the fetus also differentiates rhythmic movements when the mother walks from the more static movements when she rests in a sitting or reclining position [35]. The fetus reacts to linear accelerations caused by changes in posture when the mother moves from a sitting to a standing position or starts walking from a static position. The fetus does not show neurovegetative changes to the frequently present stimuli, which, however, does not mean that the fetus does not perceive them [36]. The rhythmic stimulation created by the mother’s footfalls is a bi-modal stimulation, both vestibular and auditory. The most salient rhythm the mother makes is when she walks, and her feet hit the ground in a repetitive and regular way. Soothing a baby by rocking (rhythmical vestibular stimulation) is common among mothers of all cultures and all ages. Mothers from various cultures (individual versus group) use rocking in the same way to soothe their children [37]. It is probably a universal human behavior whose relationship to maternal walking during pregnancy has not been investigated [38]. To the best of our knowledge, no study has shown a link between the maternal walking rhythm and the rocking rhythm of the newborn child.

#### 1.1.5. Intersensory Redundancy

Animal and human research over the past 30 years has shown that intersensory redundancy promotes attention, learning, and memory for modal stimuli, such as tempo, rhythm, and intensity [39]. In humans, the mother’s singing makes her spine vibrate, synchronized with movements of her diaphragm, and is often accompanied by movements of her body. This intersensory redundancy, i.e., when the same information (maternal singing) is available simultaneously and synchronized between at least two (vestibular and auditory) sensory systems (diaphragm movement and the sound), allows the fetus to better perceive and process the information. The tempo of the mother’s walk is temporally synchronized with the sound of her footsteps and the tactile feedback as the fetus experiences changing pressure as well as the accompanying and coordinated vestibular changes caused by the mother’s movements. After birth, this intersensory redundancy is particularly important during social exchanges: for example, the tempo of a speech can be perceived both by listening and by looking at the interlocutor’s mouth. This redundancy also occurs when the mother carries her child in her arms while singing and rocking the child to the same rhythm. Sensory redundancy can contribute to the emergence and development of early postnatal social motivations [4]. Multimodal stimulation has neurological effects that consistently exceed the level predicted by the addition of each separate unimodal stimulus. This underscores the importance of multimodal information in facilitating selective attention and perceptual learning in early childhood [40]. Synchrony, intensity, rhythm, and tempo (information that is common across the senses) are amodal information that can be detected by the fetus and infants through multimodal redundancy across the sensory system and facilitate prenatal learning [40]. Such temporally synchronized, redundant prenatal sensory stimulation can facilitate the development of neonatal social motivation, social recognition, social learning, and memory [41].

#### 1.1.6. Links between Multimodal Fetal Rhythms and Music

Iversen [3] hypothesized that because of the link between heartbeat, breathing, vocalization, and locomotion, the walking pace might be the rate that humans prefer to synchronize. The multimodal perception of sound and vestibular stimulation caused by the maternal walk would be at the origin of the connection between dance and music [10]. All music has a rhythm and a beat. Music is often played at a tempo similar to that of walking [42] because the average speed of a rhythmic pulse or of music, called moderate tempo (neither fast nor slow), is around 100 beats/min, or 600 ms/beat [43], which is also the spontaneous motor tempo (SMT) of an adult who is asked to tap with fingers on a table as regularly as possible at a rhythm that seems as natural as possible [44]. Research on experimental psychology and cognitive neuroscience indicates that rhythm and movement are tightly linked [45]. Almost all musical rhythms naturally induce human movement. Rhythmic perception depends on body movement [38]. Rhythm is indispensable for both dance and music [46].

### 1.2. Rhythm Production In Utero

A fetus is also able to produce different rhythmic patterns: cardiac pulsations, breathing movements, hiccups, sucking, arm and leg movements, and even crying [47]. All these rhythmic patterns have a spontaneous motor tempo (SMT); for a review, see [2]. Van Leeuwen et al. [16] suggested how the in utero fetal heart rate may synchronize with external rhythmic stimuli from the mother. The fetus’s ability to adjust its cardiac rate to the external rhythmic stimulation by the mother is observed as early as the third trimester of pregnancy [16]. At birth, motor production is influenced by rhythmic stimulation, whether auditory or audiovisual rhythmical stimulation. The newborn is able to modify the rhythms of some of its activities (non-nutritive sucking, vocalization) in order to be synchronized with the rhythms of its environment [2,48]. The adjustment is finer when the rhythmic stimulation is close to the SMT. It is also easier to accelerate the SMT than to slow it down, even with adult speech [49]. A newborn’s vocalizations are more easily synchronized to rhythmic stimulation than to motor behavior. In addition, during the interaction between mother and newborn, the latter does not vocalize at any time during the exchange. Most of its vocalizations occur 50 ms after the end of the mother’s vocalization. A newborn’s vocalizations are synchronized to the mother’s vocalizations [50]. These results support the idea that the synchronization of movement may promote prosocial behavior [51]. The development of rhythmic abilities is, therefore, linked to the first social interactions and the development of social abilities. The perception of temporal regularities as well as synchronization abilities favors the development of these socio-cognitive skills [52,53]. The ability to synchronize is the critical developmental milestone that allows a toddler to interact in the mother–infant dyad and then with multiple people [3].

### 1.3. Rhythmic Stimulation in NICU

The intrauterine environment is rhythmic, but such rhythm is completely absent for the preterm infant in the incubator. The NICU environment deprives infants of sensory stimulation [54]. Lahav and Skoe [55] described the complex sound environment in the womb as rhythmic, periodic, organized, and predictable, while in the NICU, the sound environment is described as aperiodic (white noise), unorganized, and unpredictable (alarms). In the incubator, the child no longer hears the heartbeat or the breathing rhythm of its mother. Regarding rhythmic language stimulation, only 2% to 5% of the sounds reaching the ears of the preterm infant are language [56]. The child remains mostly lying in a horizontal position and has no vestibular stimulation and even less vestibular rhythmic stimulation. Stationary confining incubators reduce the amount of vestibular information available to the infant [57]. However, if the mother is bedridden, a young preterm infant may be deprived of prevalent in utero rhythmic vestibular stimulation [58]. In general, preterm infants receive significantly less rhythmic vestibular stimulation related to walking than fetuses of the same gestational age, who receive stimulation from an average of 5000 maternal steps per day [55]. In fact, in the incubator, the preterm infant lies mostly on its back (or on cushions in an appropriate NIDCAP method), and because of all the wires that connect the infant to devices, it does not receive any walking vestibular stimulation.

#### 1.3.1. Rhythmic Vestibular Stimulation

All these factors in the NICU environment have a clear impact on the development of the infant [6]. To the best of our knowledge, no study has investigated the effects of restricted rhythmic vestibular stimulation on the global development of the preterm infant. As the vestibular system is one of the first sensory systems to develop (from a gestational age of 15 weeks), it is the sensory system that should provide the most appropriate developmental stimulation to the preterm infant [54]. Rocking generates vestibular stimulation that has the same rhythm as the maternal walk and consequently the movements of the child in utero [58]. Because of the universally calming effect that rhythmic vestibular stimulation has on the newborn, many researchers have developed rocking stimulations in neonatal care units over the past 50 years. Richter and Ostovar [38] identified 157 publications on rocking in the medical literature between 1948 and 2014. In a systematic review of 15 articles published between 1970 and 2017 on the effect of manual and mechanical tactile stimulation on apnea in preterm infants, Cramer et al. [59] found seven articles on rhythmical stimulations, essentially with water mattresses. Korner et al. used waterbeds to reduce apnea in preterm infants. They used a water-filled mattress with irregular head-to-foot oscillations of 12 to 14/min, giving vestibulo-proprioceptive stimulation similar to that in utero. They reported reduced apnea in preterm infants. Subsequently, the same team continued to use waterbeds, but with irregular rhythmic stimulation frequencies (ranging from 8 to 16 oscillations/min) [60,61]. These authors did not always find beneficial effects in terms of reducing apnea but found an increase in sleep time during which apnea and bradycardia were reduced. Groswasser et al. [62] proposed continuous body rocking with the use of an inflated mattress. Side-to-side rocking was obtained at a speed of 13 cycles/min for one night. In preterm infants prone to apnea, side-to-side body rocking is associated with a significant decrease in the frequency of obstructive breathing events during sleep, which is not the case when the rocking is too regular, without variability [63], or when the mattress is in air with too regular oscillations of 14 to 16 cycles/min [64]. A rocker fitted into the incubator and rocked gently for two 30 min periods daily induced more smiling and no signs of tension in preterm infants. It appears that the rocker induced states of hyper-relaxation [65]. Preterm infants exposed to 15 min sessions of sinusoidal oscillation rocking three times a day for 2 weeks showed significant gains in motor skills and overall neuromuscular maturation compared with the controls [66]. Rhythmic stimulation, vestibular and/or auditory, has also been used to successfully train the respiratory rhythm of preterm infants. This has immediate and important implications for their health status [67]. The authors noted that the preferred rocking rhythm for synchronization with the respiratory rhythm is 42–50 cycles/min. If the rocking rhythm is faster, above 50 cycles/min, it does not synchronize with the respiratory rhythm. If the rocking is slower, 32–40 cycles/min, the preterm infant breathes twice during a single rocking. Again, as with newborns, rhythmic synchronization can only occur when the rhythm of the vestibular stimulation is close to the child’s spontaneous motor tempo. Tuck et al. [68] constructed a rocking bed that imparts a regular cephalocaudal rocking movement. The rate of rocking, constant for everyone, ranged from 10 to 22 (mean 16.5) cycles/min. Preterm infants had less apnea when the bed was rocking than when it was still. With the bedside device developed by [54], Barlow et al. [69] provided to preterm infants, seven different rocking stimulations (linear horizontal motion stimuli) that vary in rate were studied, showing that preterm infants can increase their respiratory rates, while maintaining a stable pulse in response to a specific rocking stimulation near their SMT [54]. This calming effect is common to all ages because, even in adults, using a rocking bed during an entire night of sleep promotes deep sleep and memory consolidation [70].

#### 1.3.2. Rhythmic Breathing Stimulation

In a 2-week intervention period, preterm infants spent more time in quiet sleep when in contact with their breathing bears than infants given non-breathing bears or the no-bear group [71,72]. The authors wired the teddy bear’s sinusoidal oscillations to the infant’s individual rhythm by taking a cadence that was half the infant’s breathing rate. The results suggest that a preterm infant can be trained by the teddy bear to breathe more regularly, always considering the child’s own SMT.

#### 1.3.3. Rhythmic Sucking Stimulation

Song et al. [73] used a pacifier (developed by Barlow et al. [69]) that delivers consistent patterned and frequency-modulated oro-somatosensory stimulation through a pneumatically pulsed pacifier interface. The pneumatic stimulator generates a series of pulses patterned as six cycles/burst followed by a 2 s pause, which transforms the pacifier into a pulsating nipple that stimulates oral facial nerves. The training session lasted 2 min and consisted of three 3 min pulsating nipple epochs and two 5.5 min non-stimulation epochs. Although infants are not orally fed milk during non-nutritive interventions, oral stimulation may increase saliva production and swallowing practice, which may facilitate synchrony between swallowing and breathing. These involvements have been shown to have multiple beneficial effects on feeding development. Compared to the control group, the experimental group showed a significant reduction in the time needed to transition from gavage to full oral feeds and in the length of stay in the NICU. The training session with a rhythmical pulsating nipple allows preterm infants to practice non-nutritive rhythmic sucking and to mature more quickly.

#### 1.3.4. Rhythmic Multimodal Stimulation

Vestibular and heartbeat sound: Vestibular rhythmic stimulation shows different effects, depending on the stimuli, the rhythms chosen, the frequency of the stimuli, the quantity of stimuli proposed, and the duration of intervention. Always seeking to be as close as possible to the stimulations provided in utero, many authors have proposed multi-modal rhythmic stimulations by combining, for example, rocking with intrauterine cardiac noise and female voice [74] or a rocker bed with a heartbeat sound, whether or not it is triggered by the preterm infant [75]. The rhythmical stimulation could take place for 15 min/h (fixed-interval stimulation group); the infant provoked 15 min of rhythmic stimulation each time it was motorically inactive for 90 s (self-activating stimulation group), or the infant provoked 15 min of rhythmic stimulation by being inactive for 90 s, but the stimulation took place only once per hour for the fixed-interval stimulation group (quasi self-activating stimulation group). As in the original study [76], compared with the control group (without any rhythmical stimulation), the immediate effect of the rhythmical stimulation was an increase in quietness in the three experimental groups, with fewer abnormal reflexes and better orienting responses. The authors highlighted the importance of contingency (that the infant can produce an action, such as triggering the rhythmic movement of the bed) and the temporal pattern of stimulation (that it occurs once every hour).Multisensory stimulations, including rhythmical stimulation: Interventions of two programs incorporate rhythmical stimulations in a multisensory stimulation program.

Auditory, tactile, visual, and vestibular (ATVV) intervention is a multimodal sensory stimulation intervention for preterm infants to improve mother–preterm interaction [77]. This intervention incorporates not only rhythmical stimulation but also eye-to-eye contact when the infant is alert and talking to the adult, light stroking or massage of the infant for the first 10 min of the interaction, and rhythmical vestibular or a slow rocking motion (horizontal rocking) while attempting eye-to-eye contact, and maintaining of auditory contact for the remaining 5 min [78]. Generally, ATVV increases the period of alertness, shortens hospital stays, enhances maternal–infant interaction (when provided by the infant’s mother), and enhances behavioral organization at term age [79,80,81,82,83,84,85,86,87,88]. This method highlights the effectiveness of multimodal stimuli, incorporating rhythm, when they are in synchrony with the infant’s reactions.

The supporting and enhancing NICU sensory experiences (SENSE) program includes skin-to-skin care, infant massage, auditory exposure (human speech, music), olfactory exposure (maternal scent, close maternal contact), kinesthetic/vestibular exposure (holding movement, rocking), and visual (dim or cycled light) exposure. The program also includes parental education fostering an understanding of individualizing care related to infant behavioral signs. Vestibular interventions include rocking for a minimum of 7 min by term-equivalent age [88]. The SENSE program is intended to increase maternal confidence in addition to bettering infant neurobehavior with less asymmetry on the NICU Network Neurobehavioral Scale (NNNS) and better Hammersmith Neonatal Neurological Evaluation (HNNE) scores. Since these are multimodal stimuli that are not all rhythmic, we cannot know how many of the beneficial effects of these stimuli are due to rhythmic stimuli, but it is still important to note them.

Mother’s voice and heartbeat sound: Studies have stimulated preterm infants with the mother’s voice and heartbeat at 30 min intervals, four times per 24 h [89]. Rhythmical maternal sound stimulation (MSS) starts within 7 days after birth and is continued until discharge from the NICU. Maternal sound and heartbeat were recorded for each infant during maternal speaking, reading, and singing. The authors found an overall decreasing trend in cardiorespiratory events (CREs) with age. With nearly the same stimulation (audio-recording of the mother’s voice and heartbeat sounds, four times per day for a duration of 45 min each over a period of 1 month), Webb et al. [90] used cranial ultrasonography measurements at 30 days of life. Extremely preterm infants exposed to their mothers’ voices and heartbeats during their first month of life in an incubator had significantly larger auditory cortexes bilaterally compared with control preterm infants, although the mother’s actual voice and live heartbeat would be even more effective [91].Music: Music interventions are especially difficult to fully describe due to the complexity of music stimuli (rhythm, pitch, tempo, harmonic, structure, timbre, jitter, shimmer, etc.), variety of music experiences, and factors due to music interventions. It is therefore difficult to know the effects of the rhythm itself. We only know that there is no music without rhythm, whereas there can be music without melody [42]. Music has often been effectively used in neonatal intensive care units, especially with high-risk infants [14]. Moreover, music is thought to improve neurodevelopment in preterm infants by promoting synaptic plasticity and the differentiation, activation, readjustment, and growth of neurons [92]. A review of music therapy in the NICU between 1970 and 2010 revealed previously unsuspected perceptual, adaptative, and active engagement capacities of preterm infants during music therapy [93]. The authors focused on music or auditory stimulation interventions that incorporated musical elements, such as rhythm and sounds, based on the acoustic rhythmic intrauterine environment, such as recorded womb sounds, the mother’s voice, breathing sounds, and heartbeats. The review showed that music has positive effects on the preterm infant, calming and relaxing the infant and decreasing its stress level. Another systematic review of music-based intervention research published from 2010 to 2015 showed poor quality of music intervention studies [94]. The authors recommended improving the reporting quality, scientific rigor, and clinical relevance of music intervention research and suggested a seven-component checklist to advance the scientific rigor and clinical relevance of music intervention research. A recent study showed that preterm infants can learn and memorize from their auditory environment and that they can discriminate music played in the neonatal unit from the same music with a faster tempo [18]. Preterm infants are therefore able to recognize the temporal structure of a known piece of music at a specific tempo and to differentiate it from the same piece played at a faster tempo. Rhythm processing has further been shown to be especially important for language processing and recognition. Early postnatal music intervention increases neural responses related to music tempo processing and recognition [95].Voice: Similar to the fetus and the full-term newborn, the preterm infant reacts more to its mother’s voice by displaying accelerated cardiac rhythm compared to when the voice is absent [96]. Just as the contingent voice is important to the infant’s responses [91], better self-regulation of the preterm infant has been observed during the interaction when the song is contingent to the infant’s reactions [97]. Similarly, the beneficial effects of singing are greater when the parents sing directly to the child versus when the mother sings as if her child were present [98]. Linguistic research shows that lullabies of all cultures combine language information and use calming, rhythmic stimuli. Lullabies, with no tempo change, were used to reinforce non-nutritive sucking rates of preterm infants. Contingent lullabies, such as pacifier-activated lullabies (PALs), increase pacifier-sucking rates of preterm infants [99], increase subsequent feeding rates [100], and shorten gavage feeding lengths when used at the specific gestation age of 34 weeks [101]. Rhythmic lullabies reinforce the sucking rates produced by preterm infants. Consequently, sucking rhythms are modified by lullabies: the more the preterm infant sucks, the more the lullabies provided. The preterm infant can learn to suck–swallow–breathe with music contingency.Kangaroo care: The importance of the multiplicity of rhythms in synchrony with each other and their possible link with musical rhythms has been shown in utero [10]. In the neonatal unit, the different rhythms are clearly less numerous and are only rarely presented together. The rare moment when the infant is again simultaneously in the presence of several rhythms in synchrony is when it is exposed to kangaroo care on the mother’s chest (skin-to-skin contact). The full-body contact and the sound of the mother’s heartbeat are thought to simulate sensations that the infant experienced prenatally [102]. In this position, the infant can again hear its mother’s heartbeat, perceive her breathing rhythm, and hear the rhythm of her speech if she is speaking or the rhythm of the song if she is singing. Intersensory redundancy is again present in a skin-to-skin-contact situation. When a mother speaks to her child (infant-directed speech), she uses the motherese, which accentuates the melodic contours and uses a slower rhythm, better perceived by the child [103]. Similarly, the infant-directed singing, used by the mother when she sings to her child, has more accentuated melodic contours and a slower rhythm that is better perceived by the child [104]. Parent–infant skin-to-skin contact, commonly known as kangaroo care, underscores the importance of maternal body contact for the infant’s physiological, emotional, and cognitive regulatory capacities [105]. Compared with kangaroo care alone, combining kangaroo care and maternal singing can be especially beneficial for mothers as it reduces their anxiety levels [106]. Here, the mother was instructed to sing with a repetitive, soothing tone, softly, simply, and with a slow tempo, i.e., the characteristics of infant-directed singing. In the preterm infants in the group exposed to kangaroo care and maternal singing, the authors observed better autonomic stability and a calming effect. During kangaroo care, the skin-to-skin contact between mother and preterm infant provides multisensory rhythmic stimulation in a unique, interactive way that can significantly decrease or mask the harmful effects of environmental stimuli. Roa and Ettenberger [107] studied kangaroo care using the rhythm, breath, and lullaby (RBL) model developed by Loewy [108] in order to replicate the auditory environment in the womb, such as slow tempo and repetition. With RBL, parents experience less anxiety, decreased stress levels, increased maternal relaxation, and more motivation [109,110] The music, the humming, and the vibration of a monochord placed on the kangaroo parent’s elbow so that the rhythmical vibrations can be felt by the preterm infant create a sense of closeness and intimacy, a new way of meeting and being together [111]. Kostilainen et al. [112] investigated the effects of daily singing combined with kangaroo carrying during the first weeks after preterm birth. Parents were encouraged to sing or hum at a slow tempo with repetitive and simple melodies during the kangaroo care for the time they liked. Parents who sang felt a positive impact on their well-being: singing improved interaction and made it easier for them to connect naturally with their child. They felt more relaxed when they sang, and they also felt that their child was more relaxed. Thus, singing during kangaroo care was mostly experienced as a shared, intimate moment between parent and infant. Can the multiplicity of rhythms created by skin-to-skin contact and the addition of a lullaby promote trans-natal continuity, potentially affording additional synchronization cues?

## 2. Empirical Pilot Study

The finding that compared with skin-to-skin care alone, combining skin-to-skin contact with maternal singing is more beneficial for mothers as it reduces their anxiety levels [106] is particularly important because of the adverse consequences of anxiety on caregiving behaviors. Based on this knowledge, we studied whether singing during skin-to-skin contact allows the mother to better focus on the premature infant and improves the mother’s caregiving behaviors in the NICU. In general, lullabies are characterized by a simple melodic structure, a slow tempo, ritardando, rallentando, repetition of words and syllables, and rhymes that are, most of the time, produced to soothe and regulate the state of a young infant. We observed whether on exposure to a combination of the rhythm of a lullaby and skin-to-skin contact, the premature infant shows less disorganized behavior. Therefore, our goal was to find out whether asking the mother to sing a lullaby to her preterm infant would result in better maternal involvement and consequently improve caregiving behavior. Since there are many in utero rhythms (as detailed in the theoretical section), would the new rhythm created by the lullaby further enhance the benefits of skin-to-skin contact?

Our pilot study aimed to assess whether the rhythmicity of the lullaby addressed to an infant improves attachment-relevant interactive behaviors during one skin-to-skin session. It is expected that a mother who sings a lullaby to her preterm infant will be more engaged, thus promoting attachment-relevant interactional behaviors.

### 2.1. Method

#### 2.1.1. Participants

In this observational study, conducted in a neonatal unit, 10 mother–preterm infant dyads participated. Preterm infants born between 32 and 37 weeks (SA) were observed within 7 days of birth. The 10 dyads were randomly assigned to one of the two groups (*n* = 5 dyads each), the lullaby group and the no-lullaby (control) group. The mothers in the lullaby group sang (infant-directed singing) during the skin-to-skin-contact session, while the mothers in the control group underwent a skin-to-skin-contact session without singing (the mothers were not given any particular instructions; they continued skin-to-skin contact in the same natural way as the first 5 min). 

The daily presence of the mother with her baby was an inclusion criterion. Mothers with psychiatric pathologies, those with hearing and/or visual deficiencies, and those who did not understand French and preterm infants with severe neurological or digestive pathologies and those on assisted ventilation (except those with oxygen goggles) were excluded from the study. The management of the NICU approved the study, and the parents signed a consent form after being informed about the conditions of the experiment. 

#### 2.1.2. Procedure 

During a calm, waking state of the preterm infant [113], skin-to-skin contact was established for each dyad. The infant’s setup was inspired by the sustained diagonal flexion position [114], with a support band and a nursing pillow to support the mother’s support arm. The preterm infant was effectively curled up against the mother’s chest and held in the crook of the mother’s arm without the mother being able to perform vestibular rocking. 

Two periods were videorecorded for both groups. In the first 5 min (baseline or initial phase), the mother–infant dyads were in classic skin-to-skin contact, ensuring that the skin-to-skin contact was correct, the mother was comfortable, and the infant was calm. In the next 5 min (test phase), the mothers in the lullaby group, still in skin-to-skin contact with the infants, sang a familiar lullaby softly and continuously. The lullabies produced spontaneously by the mothers of the lullaby group were all traditional French songs such as “fais dodo Colas mon p’tit frère”, which may be sung at different tempi depending on context (e.g., infant state), and were sung during the whole five minutes. In the control group, the mothers just continued the skin-to-skin session with the infants in the same natural way, without singing. Simultaneously, the cardiorespiratory parameters of the infants (number of bradycardia and oxygen desaturation events) were recorded. Each mother performed a single skin-to skin session (for at least 10 min); see Figure 1.

#### 2.1.3. Data Analysis

The behavioral indicators of mothers and infants were analyzed in terms of duration and frequency. For the mother, two behaviors highlighting the implementation of caregiving were retained. The gestures of the mother’s active hand (the other hand supporting the infant) were classified as hand in contact with the infant (caressing or wrapping) and hand without contact. The gaze was classified as follows: the mother looks at her baby or the mother looks away.

Concerning the preterm infants’ manifestations, two main behavioral indicators of the progressive establishment of attachment were analyzed. The gaze was classified as eyes closed or a disorganized gaze, and facial expressions were classified as a smile (upward movement of the lips with squinting of the eyes), grimace (random and fluctuating contractions of the facial features), or neutral face (no movement). Due to the physiological immaturity of the preterm infants, their gazes and smiles could not be observed for a sufficiently long period to obtain useful results.

During video analysis, for both phases (initial and test) and for each appearance of a given behavioral item, start and end time markers were noted in order to obtain the time interval, i.e., the duration of each item (unit duration). Each item could appear several times for 5 min; the total duration (or cumulative duration) was obtained by adding these different unit durations. This total duration (cumulative duration) did not indicate the length of time each item unit lasted each time it occurred. Thus, the average duration (total duration divided by the frequency) was also calculated in order to determine whether each item (on average) was maintained when it appeared.

### 2.2. Results

Statistical analyses for the two independent groups (with small numbers) were carried out using the non-parametric Mann–Whitney U test (significant at *p* < 0.05). 

In the initial phase, a comparison of physiological constants allowed us to certify that the two groups were comparable. A comparison of physiological constants between the preterm infants in the two groups showed an equivalent number of oxygen desaturation and bradycardia events. Comparisons of the cumulative and unit durations of each item observed in the mothers and preterm infants showed no significant difference, highlighting the behavioral equivalence of the two groups. 

In the test phase, a comparison of physiological constants between the preterm infants in the two groups showed an equivalent number of oxygen desaturation and bradycardia events, indicating the absence of a deleterious effect of the lullaby.

There were no significant differences between the 5 min of the initial phase and the 5 min of the test phase in the control group. The following main results focused on the comparison of behavior in the test phase when the mothers sang versus when they did not sing. For the test phase, we selected some behavioral items showing significant differences between the two groups.

The total duration of maternal gestures involving contact with the infants (caresses and wraps) was significantly longer in the lullaby group (mean = 277 s) compared with the control group (mean = 152.6 s) (*p* = 0.016); see Figure 2.

The average unit duration of maternal gestures involving contact with the infants was also longer (but not significantly) in the lullaby group (mean = 81.5 s) compared with the control group (mean = 21.7 s). A caress or a hand wrap by the mother lasted longer when she sang than when she did not. 

Concerning maternal gaze on the preterm infant, the total duration of the gaze was significantly longer in the lullaby group (mean = 288.6 s) compared with the control group (mean = 221.4 s) (*p* = 0.028); see Figure 3. Furthermore, the average duration showed that each gaze lasted significantly longer on average in the lullaby group (mean = 195.3 s) compared with the control group (mean = 13.7 s) (*p* = 0.016). Thus, mothers singing a lullaby sought more eye contact with their babies, presenting better gaze stability (less disruption).

Closed eyes are associated with sleep or relaxation in newborns. The total time the preterm infants spent with their eyes closed was longer in the lullaby group (mean = 279.8 s) compared with the control group (mean = 185.4 s) (*p* = 0.09 close to significance); see Figure 4. Facial expressions were more neutral, and there were fewer grimaces in infants in the lullaby group compared with the control group, but the difference was not significant. This result indicates better well-being of infants exposed to a lullaby. The total duration of disorganized gaze was significantly lower in infants in the lullaby group (mean = 1.4 s) compared with the control group (mean = 109.8 s) (*p* = 0.009). The mean unit duration lasted less in the lullaby group (mean = 1.1 s) compared with the control group (mean = 12.4 s) (*p* = 0.02). This result highlights a decrease in the discomfort and restlessness of the infants during the lullaby. 

### 2.3. Discussion

Despite the small number of participants in the study, the results point to an important contribution of the lullaby during skin-to-skin contact between mother and infant, especially in a single session. Singing stabilizes the mother’s gaze on her infant (longer duration of gaze on the infant and shorter duration of gaze elsewhere) and promotes the preterm infant’s state of relaxation (more time with eyes closed). By creating a unique, intimate, quiet bubble, stimulating hearing and mutual discovery, the lullaby enhances the effects of skin-to-skin contact in a single session and may encourage a reciprocal mother–infant interaction. Skin-to-skin contact with the mother singing a lullaby seems to offer the preterm infant more intersensory redundancy: the preterm infant hears the mother’s heartbeat and breathing rhythms and voice in synchrony with her lip movements, and the mother looks at and touches her child more. The strokes are rhythmic. When the mother is asked to stroke her child, she instinctively adopts a mean velocity rate (MVR) that matches the optimal C-tactile afferents (CTs: a class of unmyelinated nerve fibers activated by low-force, dynamic, rhythmic touch) stimulation range of 1–10 cm/s [115,116]. The period of optimal velocity rhythmic stroking produces a greater reduction in physiological arousal in such preterm infants than in those receiving static, non-CT-optimal touch [117]. The videos of our empirical pilot study did not allow us to analyze the rhythmicity of touch. Future studies could use motion capture in order to analyze rhythmic stroking and assess whether there is a higher amount of and/or better synchronized maternal caresses when the mother sings a lullaby than when she does not. This multiplicity of rhythms in synchrony seems beneficial for a preterm infant. It would be interesting to assess whether the mother’s caresses are also more synchronized to the newborn’s contingency responses [50,118] when she sings a lullaby.

## 3. Conclusions

In the intrauterine environment, the fetus is exposed to multiple patterns of repetitive vestibular, tactile, somatosensory, auditory, and visual stimulations. Many of these stimulations are rhythmic, and the fetus detects these patterns and differentiates their variable rhythms. The fetus recognizes these different rhythms (the mother’s heartbeat, breathing, voice, walking, etc.) and modifies its own behavior in response to the mother’s rhythms. This paper highlighted the rhythms perceived by the fetus in the intrauterine environment and the lack of these stimulations in the incubator. In the NICU, rhythmical stimulation is missing. The various sensory and rhythmic interventions offered by the mother have beneficial effects on development, considering the preterm infant’s behavioral states. Therefore, individualized rhythmic stimulation that is contingent on the responses of the preterm infant should be favored. When the stimulation is not contingent, the infant is emotionally disturbed [119]. These rhythmic stimulations in everyday life have a natural variability that allows the synchronization of the rhythms of the parent–infant dyad and facilitates selective attention and perceptual learning [120]. Among all the interventions, the classic method of skin-to-skin contact offers the preterm infant the possibility of simultaneously receiving, in synchrony, all the maternal rhythmic signals. Coupled with maternal singing, skin-to-skin contact seems to better stabilize the physiological constants of the preterm infant and favors closeness in the mother–infant dyad. When the mother sings to her child in the kangaroo care, she offers the child a multitude of rhythms (respiratory, cardiac, singing). Skin-to-skin contact along with infant-directed singing generates a rhythmical synchronization between mother and infant, providing an envelope (tuning) of several rhythmic stimuli [121]. By offering the preterm infant the multiplicity of rhythm that they had in utero, the caregiver improves the interaction, and the infant may be engaged in rhythmic synchronization [122] and interactional synchrony [105]. When it becomes possible to have infants connected to wireless/portable connection, it would be interesting to study the influence of skin-to-skin contact along with singing while the mother is walking. The superposition of this vestibular rhythmic stimulation of walking would positively complete the multiple rhythmic stimulations that the infant could receive that are in synchrony with one another and consequently in synchrony with the mother. Synchronous interactions, in which both parent and preterm infant are mutually responsive, are also important for developing attachment [123]. When the preterm infant and the caregiver are in synchrony, it greatly enhances the well-being of the parent–infant dyad and plays a central role in the infant’s emotional, social, and cognitive development.

## Figures and Tables

**Figure 1 children-08-00660-f001:**
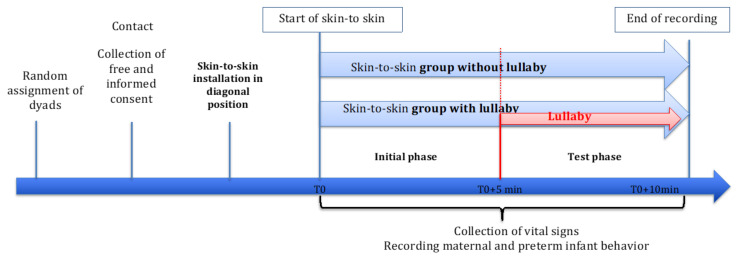
Study protocol (T0: Start of skin-to-skin/recording).

**Figure 2 children-08-00660-f002:**
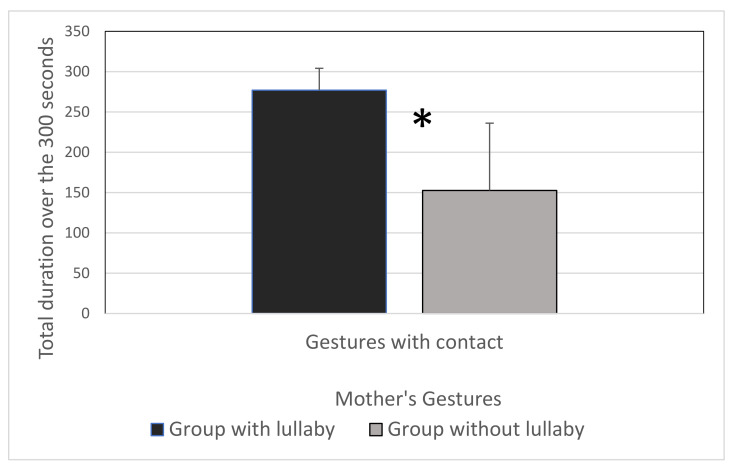
Total duration of gestures involving contact of the mother’s active hand for the lullaby group and the no-lullaby group (* *p* < 0.05).

**Figure 3 children-08-00660-f003:**
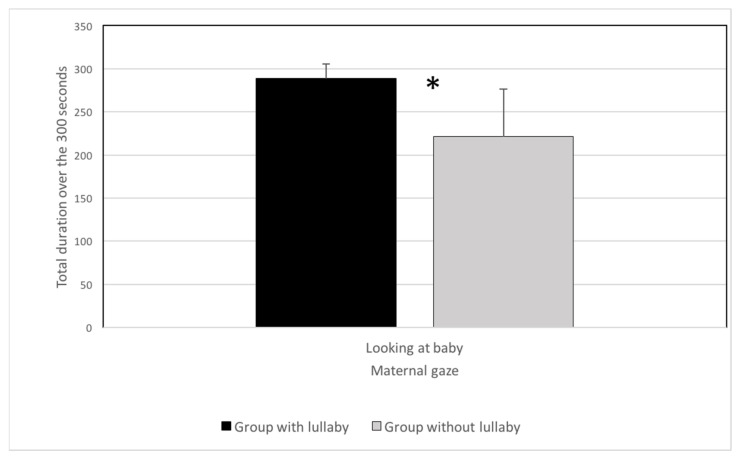
Total duration of the maternal gaze on preterm infants in the lullaby group and the no-lullaby group (* *p* < 0.05).

**Figure 4 children-08-00660-f004:**
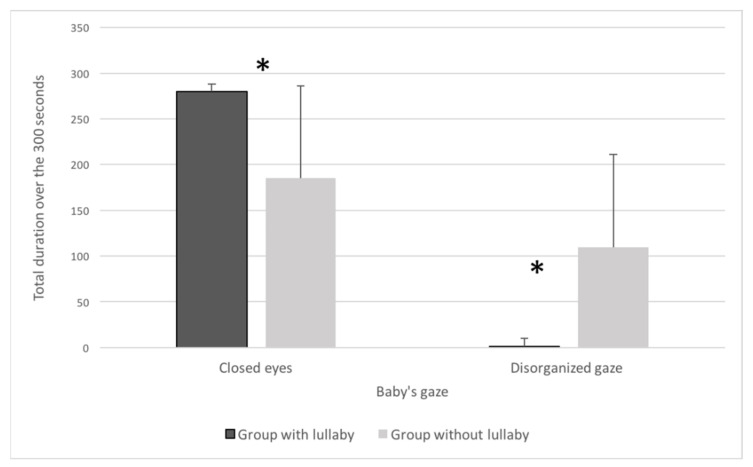
Total duration of closed eyes and disorganized gaze of the indicator “baby’s gaze” for the lullaby group and the no-lullaby group (* *p* < 0.05).

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
