# Peer review of "The Importance of Rhythmic Stimulation for Preterm Infants in the NICU"

_children, 2021, doi:10.3390/children8080660_

Round 1

Reviewer 1 Report

This manuscript provides an overview of rhythmic experiences and responses for the human fetus and premature newborn, then describes an observational study of lullaby singing for preterm infants in a neonatal unit.

The article provides some helpful, encouraging information, yet requires significant revision to make a substantial and innovative contribution to existing knowledge regarding infants, rhythm, and music.  To that end, I offer my constructive comments, which I hope are helpful, and I do encourage you to continue pursuing this important work. 

From an overarching perspective, I have two concerns about the manuscript:

  1. It appears to be two different papers. The first 6-7 pages specifically address rhythm and infants, while the observational study section expands considerably into lullabies for infants.  As a musical form, lullabies encompass far more musical elements than rhythm (i.e., melody, harmony, timbre, jitter, shimmer, etc.), and in fact may lack a strong rhythmic pulse or sense of rhythmic repetition.  The manuscript needs revision to be overall more cohesive, and to provide a stronger justification for the use of lullabies for preterm infants (i.e., is the effect of the lullaby due to rhythm, or due to other factors?).  I do believe that lullabies (and other song types) can be beneficial for preterm infants, but in its current form, the manuscript needs to better and more fully explain this benefit – beyond rhythm. 

  1. The information on rhythm and infants requires significantly greater clarity, expansion, and depth in order to be meaningful and useful to the reader. I do believe that rhythmic input can help preterm infants organize their behavior.  However, the literature reviewed seems rather formulaic in simply describing the rhythmic input given and resulting changes in infant behavior.  In order to best apply rhythm in the NICU, one needs to understand and articulate (or at the very least propose) an underlying mechanism of change.  How is it that rhythm impacts preterm infant behavior?  For example, does rhythm elicit central nervous system or autonomic system activity that also regulates arousal states, or metabolic processes such as respiration?  To simply state that rhythm helps infants, without explaining this mechanism, could lead to misapplication of rhythm and realization of unintended or undesirable outcomes. 

I am including more specific comments below, some of which pertain more to writing style and mechanics of writing:

In the Introduction, I see valuable information, yet the writing is a bit cumbersome.  As an example:

Line 28, you write: “Knowing when to produce one's action so that it is synchronized with the action of the person who wants to interact with one has to be learned during development from a very young age.”

Could be written as:  “The ability to produce an action so that it intentionally synchronizes with another person’s action is typically learned at a young age.”

Further, you write – this ability “has” to be learned from a young age – why is this?  Is there a critical window for acquisition of this ability?  Further, what are your thoughts on the role of mirror neurons in regard to the acquisition of synchronization?

Work to reduce use of passive voice, and revise to active voice which will enhance the smoothness, conciseness, (and digestibility) of your writing.  For example:

Line 37 you write: “There is no doubt that the fetus can perceive sounds and movements….”

Try:  “The fetus can definitely perceive sounds and ..”

Or:  “Reliable evidence confirms that the fetus can …”

Line 40:  when you say “…the mother’s voice, word, (etc.) can be perceived intermittently..” do you have a reference for this statement?  I would be interested to see how is it known that the fetus can perceive such stimuli (through heart rate, brain stem response, etc.)?

Live 45 also is written in passive voice:  “However it is obvious that these low frequency maternal sounds..”

Consider:  “These low frequency maternal sounds, such as heartbeats, are obviously audible to the fetus…”

Some information needs expansion.

Line 49 you write:  “…fetus perceives changes in heart rate according to mother’s activity level and stress ..”  What kinds of changes?  And how is this known?

Line 52 you write:  “Maternal heartbeat is the fetus’ first metronome.”

What does this mean?  That the fetus somehow uses that input to regulate behavior?

Line 52-53 you write:  “Fetuses are conditioned to respond to heartbeats…”

How do fetuses respond?  What evidence exists that such responses would generalize to other rhythmic sounds?

I find section 1.1.2 Maternal breathing to be confusing; I’m not seeing the connection between breathing, heart rate, and walking rhythm.  In fact, walking rhythm is not addressed in the paragraph (but is addressed later in section 1.1.4).  Better organization and sequencing are needed.

Section 1.13 is one very long paragraph.  Consider revising/dividing to emphasize main points and enhance readability.  I find the writing needs further revision for clarity; strive to contextualize the information and provide a logical thread for the reader to follow.  Avoid “listing facts.”    

Lines 95-99, I find inter/sensory redundancy is an intriguing concept.  Your paper could be strengthened by further explaining and justifying this concept in greater detail.  Certainly it has ramifications for social motivations, but could it also contribute to nervous system development that is necessary for typical development and functioning across several domains?

In the first 3 pages, I’m seeing a lot of claims about infant and fetus behaviors and rhythm, but these responses are not specifically described (what are they?), nor are the data collection methods used to determine them.  This point is problematic for the reader who wants to absorb new information and thus may know little about your topic.

As a personal preference, I recommend not referring to a fetus or infant as “it.”  As an example:

Line 121 you write:  “The fetus reacts to linear accelerations caused by changes in posture, when its mother goes from sitting to standing…”

Try:  “The fetus reacts to linear accelerations caused by changes in posture, when the mother moves from sitting to standing…”

Section 1.14 maternal footfalls is one very long paragraph and difficult to absorb.  Again, strive to contextualize the information and provide a logical thread for the reader to follow.  Avoid “listing facts.”

Section 1.2 Rhythm production in utero

Line 151-152 what kinds of rhythmic patterns does a fetus produce?  And what is the range of their SMT?

You write:  “It is also easier to accelerate the SMT than to 160 slow it down.”

Do you mean it is easier for the mother/parent to accelerate infant SMT, than to slow it down?  Or, that it is easier for the infant to accelerate SMT?

Line 162, you write

“These results support the idea that synchronization of movement may promote prosocial behavior [45].”  I don’t yet see how you reached this conclusion; please explain.

1.3. Rhythmical stimulation in NICU (I recommend stating “rhythmic stimulation)

Starting line 169; good information here in terms of describing the restrictions of the NICU environment.

1.3.1. Rhythmical vestibular stimulations (again, try “Rhythmic vestibular stimulation)

In this section, I’m not sure why you address weight gain.  What evidence exists to suggest that rhythmic stimulation leads to weight gain?  What is the common mechanism between rhythm perception and weight gain?  This section is written as one long paragraph; needs revision and separation to be meaningful to the reader.

Line 239, you write:  “Appropriate Rhythmic vestibular stimulation close to their spontaneous motor tempo delivered to preterm infants modulates respiratory rate [48]. This calming effect is universal to all ages because, even in adults, using a rocking bed during a whole night of sleep promotes deep sleep and memory consolidation [65].”

When you say” “modulates,” do you mean increase or decrease?  I’m not sure how increasing (as an example of modulating) respiratory rate would be indicative of a calming effect.

1.3.3. Rhythmical sucking stimulations (try Rhythmic sucking stimulation)

Line 252, a critical piece seems to be missing here:  how is it that feeding development improves with the use of rhythmic stimuli as part of non-nutritive sucking?  What specifically changes in regard to the infants’ oral-motor behavior (that can then subsequently impact feeding behavior)?  And are these results specific to premature infants?  Please clarify.

Section 1.3.4

Line 273 you write:  “Kramer and Pierpont (1976) [71] showed a significant weight gain and head growth of 11 preterm infants, with 60-min sessions, 8 times/day, of mechanical rocking of the waterbed and playing of a taped simulated heartbeat (72 beats/minute) simultaneously with a female voice during the rocking period.”

I’m not sure how multimodal stimulation leads to weight gain and head growth.  What is the underlying mechanism for this effect?  And this research is very outdated; if you feel it is still relevant, please justify. 

Voice and music

Line 329 you write:  “We will not dwell on the importance of speech and music in the presence of premature infants because this must be dealt with extensively in other chapters, but…”

Is this manuscript intended to be a book chapter?

In the section on Kangaroo Care, your manuscript content begins to extend far beyond rhythm and incorporates many other musical elements/stimuli.  This expansion makes it hard to determine which element or stimulus is eliciting the desired effect.  You also have not justified musical elements other than rhythm in the preceding sections.

Line 364 you write:  “Concerning the premature children in the group with kangaroo care and maternal singing, the authors observed that changes in Heart Rate Variability (HRV) indicated better autonomic stability and a calming effect: the decreased Low Frequency/ High Frequency ratio on the preterm infant with maternal singing was related to a calming effect.”

How did HRV change?  Please be specific.  What do you mean by “Low Frequency/ High Frequency ratio” and is this an effect observed on the infants, or in the infants?

When describing Rhythm, Breath and Lullaby, you state:

“The music during intervention is typically improvised and based on elements such as slow tempo and repetition.”

What, specifically, is improvised?  Does this involve improvised singing?  Is this different than infant-directed singing?  Is the mother singing, or is the music therapist singing?

Or, does the improvisation involve improvised movement?  Or improvised playing of instruments?

You discuss the outcomes of this approach for parents – what about the infant?

And is this technique best applied to full-term or premature or medically-fragile infants?  Please clarify.

Carefully consider what you are recommending (again, to avoid mis-use).  Is a lullaby the only kind of song to sing to an infant?  The gist of infant-directed singing is to sing in a way that either reflects or beneficially modifies infant state.  Given this definition, is a lullaby always appropriate, or might other song styles also be appropriate?

When describing music therapy for preterm infants, your writing could be strengthened by reference to some of the leading international researchers in the field, notably:

Helen Shoemark

Deanna Hanson-Abromeit

Friederike Barbara Haslbeck

Lori Gooding

  1. Our exploratory study

What do you mean by “this observational study?”  Does your study have a design?

While a protocol diagram is always helpful, Figure 1 did not display clearly for me.

When describing the mothers’ behavioural indicators, you mention “the mother’s non-bearing and active hand..” What is a non-bearing hand?

Were mothers given any definition of a lullaby?  Were mothers provided with any specific instructions on how to sing?

What is meant by “tonic-emotional manifestation?”  Ideally, this term should be clearly defined in a prior section of the manuscript.

Regarding data collection, you mention:

“During the video analysis, for both phases (initial and test) and for each appearance of a given behavioural item, a start and end time marker was noted in order to obtain the time interval, i.e. the duration of each item (unit duration).”

What was the time interval?  Did you use time sampling?  Did you measure duration or frequency of behavior?  Or percentage of time engaged in a particular behavior?  Please clarify. 

Line 434 What is desaturation?

Figures 2 and 3 are helpful.  Do the asterisks indicate significant differences?  Please explain.

You appear to have used an ANOVA (F test) to analyze the data.  How is the F test appropriate with such a small sample size and just two groups? 

Is the “significance” you obtained reliable in such a small sample?   This point should be addressed as a limitation.

Line 470, you write:  “Concerning the disorganised looks at the baby, the unit duration of each disorganised look at the baby lasts less in the group with a lullaby..”

Do you mean to say “disorganized looks of the baby..?”  You are talking about infant gaze here, not the mother’s gaze toward the infant, correct?

You seem to have collected data on three infant gaze types, but only reported results for two gaze types (disorganized and eyes closed).  Results should be presented for all gaze types.

Please provide a more in-depth description of the data for infant facial expression, even if just neutral.

What was the importance or role of rhythm in the lullaby singing within your observational study?  You seem to have strayed significantly from your original point of the impact of rhythmic input on fetal and newborn well-being, as emphasized through much of the first 7 pages of the manuscript.  I don’t see yet that the review of literature is strongly related to the clinical case study.

The conclusion needs revision to better broaden the reader’s understanding of the topic and raise questions/ideas for future research or clinical applications.  Right now, the conclusion appears to be a brief summary of the manuscript. 

Some final thoughts to apply throughout the manuscript:

  • Throughout, when reviewing research findings, please clarify whether significant or non-significant (statistically or practically).
  • I see use of the words fetus, infant, toddler, and child. To avoid confusion (and again, misuse of recommendations) I recommend defining the age range(s) of interest and being consistent with terminology.
  • Many small type-o’s throughout the manuscript would be easily corrected through careful proofing or use of a good copy editor.
  • Strive for correct sentence structure and consistent verb tense throughout the manuscript.

Author Response

We would like to thank the reviewers for their insightful comments on the manuscript. Please find our responses to your different point below, in green. We modified the manuscript accordingly.

This manuscript provides an overview of rhythmic experiences and responses for the human fetus and premature newborn, then describes an observational study of lullaby singing for preterm infants in a neonatal unit.

The article provides some helpful, encouraging information, yet requires significant revision to make a substantial and innovative contribution to existing knowledge regarding infants, rhythm, and music.  To that end, I offer my constructive comments, which I hope are helpful, and I do encourage you to continue pursuing this important work. 

From an overarching perspective, I have two concerns about the manuscript:

  1. It appears to be two different papers. The first 6-7 pages specifically address rhythm and infants, while the observational study section expands considerably into lullabies for infants.  As a musical form, lullabies encompass far more musical elements than rhythm (i.e., melody, harmony, timbre, jitter, shimmer, etc.), and in fact may lack a strong rhythmic pulse or sense of rhythmic repetition.  The manuscript needs revision to be overall more cohesive, and to provide a stronger justification for the use of lullabies for preterm infants (i.e., is the effect of the lullaby due to rhythm, or due to other factors?).  I do believe that lullabies (and other song types) can be beneficial for preterm infants, but in its current form, the manuscript needs to better and more fully explain this benefit – beyond rhythm. 

  1. The information on rhythm and infants requires significantly greater clarity, expansion, and depth in order to be meaningful and useful to the reader. I do believe that rhythmic input can help preterm infants organize their behavior.  However, the literature reviewed seems rather formulaic in simply describing the rhythmic input given and resulting changes in infant behavior.  In order to best apply rhythm in the NICU, one needs to understand and articulate (or at the very least propose) an underlying mechanism of change.  How is it that rhythm impacts preterm infant behavior?  For example, does rhythm elicit central nervous system or autonomic system activity that also regulates arousal states, or metabolic processes such as respiration?  To simply state that rhythm helps infants, without explaining this mechanism, could lead to misapplication of rhythm and realization of unintended or undesirable outcomes. 

We added this sentence in the introduction (line 37): The presence of sensory stimuli is essential for the development of sensory modalities. This explains why some modalities develop earlier than others, depending on the nature and frequency of the stimuli present in the environment. They play a very important role in the initiation, consolidation, modulation, specificity of construction and functionality of neural connections. Thus, environmental stimuli orient the neuro-cognitive development of the unborn child.  And the other one line 44: there is increase in fetal cortical brain activity in response to species typical sound.

I am including more specific comments below, some of which pertain more to writing style and mechanics of writing:

In the Introduction, I see valuable information, yet the writing is a bit cumbersome.  As an example:

Line 28, you write: “Knowing when to produce one's action so that it is synchronized with the action of the person who wants to interact with one has to be learned during development from a very young age.”

Could be written as:  “The ability to produce an action so that it intentionally synchronizes with another person’s action is typically learned at a young age.”

We have changed this sentence

Further, you write – this ability “has” to be learned from a young age – why is this?  Is there a critical window for acquisition of this ability?  Further, what are your thoughts on the role of mirror neurons in regard to the acquisition of synchronization?

Work to reduce use of passive voice, and revise to active voice which will enhance the smoothness, conciseness, (and digestibility) of your writing.  For example:

Line 37 you write: “There is no doubt that the fetus can perceive sounds and movements….”

Try:  “The fetus can definitely perceive sounds and ..”

Or:  “Reliable evidence confirms that the fetus can …”

We have changed this sentence: The fetus can definitely perceive sounds ans ….

Line 40:  when you say “…the mother’s voice, word, (etc.) can be perceived intermittently..” do you have a reference for this statement?  I would be interested to see how is it known that the fetus can perceive such stimuli (through heart rate, brain stem response, etc.)?

We misspoke. By intermittent we meant that the stimulation of the mother's voice is not continuous like the heartbeat or the breathing rhythm. The mother does not talk all the time, just as she does not walk all the time either. These stimulations are not permanent but intermittent. We replace by the sentence: In addition, the mother's voice, words, songs, music, and footfalls are not emitted continuously, without interruption, like the heartbeat and breathing rate, but each time the mother speaks, sings or walks

Live 45 also is written in passive voice:  “However it is obvious that these low frequency maternal sounds..”

Consider:  “These low frequency maternal sounds, such as heartbeats, are obviously audible to the fetus…”

We changed this sentence to the one you proposed

Some information needs expansion.

Line 49 you write:  “…fetus perceives changes in heart rate according to mother’s activity level and stress ..”  What kinds of changes?  And how is this known?

We have added the requested explanation. The heart rate of the fetus increases significantly when the mother with an above-average anxiety score is asked to perform a stressful task for 5 min, whereas the heart rate of the fetus is not significantly modified when the mother with a below-average anxiety score is asked to perform the same stressful task.

Line 52 you write:  “Maternal heartbeat is the fetus’ first metronome.”

What does this mean?  That the fetus somehow uses that input to regulate behavior?

We don't understand this idea like that. For us, and for Ullal-Gupta  (2013), this first metronome could influence subsequent preferences for other periodic auditory stimuli.  We have added the following sentence: The continuous, rhythmic sound of the maternal heartbeat is the most prominent and frequently heard stimuli in utero[10].

Line 52-53 you write:  “Fetuses are conditioned to respond to heartbeats…”

How do fetuses respond?  What evidence exists that such responses would generalize to other rhythmic sounds?

The fetus can learn by means of habituation, classical conditioning, and exposure learning. Researcher observed the fetal learning by changes in the fetal heart rate, either by deceleration or acceleration. We change the sentence: Because the heartbeat is the first regular and periodic stimulus a fetus hears, it could influence subsequent preference generalized to many other slow rhythmic sounds.

I find section 1.1.2 Maternal breathing to be confusing; I’m not seeing the connection between breathing, heart rate, and walking rhythm.  In fact, walking rhythm is not addressed in the paragraph (but is addressed later in section 1.1.4).  Better organization and sequencing are needed.

We delete the sentence: Research literature has often associated the rhythm of maternal breathing with heart rate and walking rhythm, and we added two sentences. The first at the beginning of the section: Very few studies have explored the fetal’ perception of the maternal respiratory rhythm. And the other one to explain Van Leeuwen et al. results: In an attempt to demonstrate the synchronization of the fetal heart rate with the maternal heart rate, Van Leeuwen et al. (2009) [15] prove that the fetus perceives the maternal breathing rate and  is sensitive to its change in rhythm.

Section 1.13 is one very long paragraph.  Consider revising/dividing to emphasize main points and enhance readability.  I find the writing needs further revision for clarity; strive to contextualize the information and provide a logical thread for the reader to follow.  Avoid “listing facts.”   

We think this paragraph is very important. We have reorganized this paragraph by changing the order of some of the sentences to make the sequence more logical and we have removed the inter/sensory redundancy to create a new section

Lines 95-99, I find inter/sensory redundancy is an intriguing concept.  Your paper could be strengthened by further explaining and justifying this concept in greater detail.  Certainly it has ramifications for social motivations, but could it also contribute to nervous system development that is necessary for typical development and functioning across several domains?

We added a section 1.1.5. Inter/sensory redundancy and we have added greater detail from the literature on the fetal development of animals: Multimodal stimulation has neurological effects that consistently exceed the level predicted by the addition of each separate unimodal stimulus. This underscores the importance of multimodal information in facilitating selective attention and perceptual learning in early childhood [38]. Synchrony, intensity, rhythm, and tempo (information that is common across the senses) are amodal information that can be detected by the fetus and infants through multimodal redundancy across sensory system and facilitate prenatal learning [38]

In the first 3 pages, I’m seeing a lot of claims about infant and fetus behaviors and rhythm, but these responses are not specifically described (what are they?), nor are the data collection methods used to determine them.  This point is problematic for the reader who wants to absorb new information and thus may know little about your topic.

 We have added details on the types of learning (habituation, classical conditioning, operant conditioning) and on the variables observed (mainly the fetal’ heart rate) without making the text too heavy

As a personal preference, I recommend not referring to a fetus or infant as “it.”  As an example:

Line 121 you write:  “The fetus reacts to linear accelerations caused by changes in posture, when its mother goes from sitting to standing…”

Try:  “The fetus reacts to linear accelerations caused by changes in posture, when the mother moves from sitting to standing…”

 We have changed for the sentence proposed

Section 1.14 maternal footfalls is one very long paragraph and difficult to absorb.  Again, strive to contextualize the information and provide a logical thread for the reader to follow.  Avoid “listing facts.”

The paragraph has been reduced and parts have been reintroduced in the redundancy section and another session has been created: links between fetal rhythms and music

Section 1.2 Rhythm production in utero

Line 151-152 what kinds of rhythmic patterns does a fetus produce?  And what is the range of their SMT?

We added the sentence: cardiac pulsations, Breathing movements, hiccups, sucking, arm and leg movements and even crying [39]

You write:  “It is also easier to accelerate the SMT than to 160 slow it down.”

Do you mean it is easier for the mother/parent to accelerate infant SMT, than to slow it down?  Or, that it is easier for the infant to accelerate SMT?

Both of all. We added a reference, Condon & Sanders1974 : When parents speed up the pace of their interaction, the newborn is always synchronized with the speech, but when they slow it down, the newborn is no more synchronized and shows dissatisfaction

Line 162, you write

“These results support the idea that synchronization of movement may promote prosocial behavior [45].”  I don’t yet see how you reached this conclusion; please explain.

We added a sentence to explain: In addition, during the interaction between the mother and her newborn, the latter does not vocalize at any time during the exchange. Most of its vocalizations occur 50 ms after the end of the mother's vocalization. Newborn vocalization are synchronized to mother vocalizations

1.3. Rhythmical stimulation in NICU (I recommend stating “rhythmic stimulation)

Starting line 169; good information here in terms of describing the restrictions of the NICU environment.

1.3.1. Rhythmical vestibular stimulations (again, try “Rhythmic vestibular stimulation)

In this section, I’m not sure why you address weight gain.  What evidence exists to suggest that rhythmic stimulation leads to weight gain?  What is the common mechanism between rhythm perception and weight gain?  This section is written as one long paragraph; needs revision and separation to be meaningful to the reader.

 The paragraph has been simplified; details have been removed to improve the reading

Line 239, you write:  “Appropriate Rhythmic vestibular stimulation close to their spontaneous motor tempo delivered to preterm infants modulates respiratory rate [48]. This calming effect is universal to all ages because, even in adults, using a rocking bed during a whole night of sleep promotes deep sleep and memory consolidation [65].”

When you say” “modulates,” do you mean increase or decrease?  I’m not sure how increasing (as an example of modulating) respiratory rate would be indicative of a calming effect.

Yes, Zimmerman and Barlow (2012) observed an increase of respiratory rate in response to vestibular stimulus. We have changed modulate by increase while maintaining stable pulse. We also made a sentence instead of the two sentences in a row.

1.3.3. Rhythmical sucking stimulations (try Rhythmic sucking stimulation)

Line 252, a critical piece seems to be missing here:  how is it that feeding development improves with the use of rhythmic stimuli as part of non-nutritive sucking?  What specifically changes in regard to the infants’ oral-motor behavior (that can then subsequently impact feeding behavior)?  And are these results specific to premature infants?  Please clarify.

We added the sentence: oral stimulation may increase saliva production and swallowing practice, which may facilitate synchrony between swallowing and breathing.

Section 1.3.4

Line 273 you write:  “Kramer and Pierpont (1976) [71] showed a significant weight gain and head growth of 11 preterm infants, with 60-min sessions, 8 times/day, of mechanical rocking of the waterbed and playing of a taped simulated heartbeat (72 beats/minute) simultaneously with a female voice during the rocking period.”

I’m not sure how multimodal stimulation leads to weight gain and head growth.  What is the underlying mechanism for this effect?  And this research is very outdated; if you feel it is still relevant, please justify. 

We deleted this sentence

Voice and music

Line 329 you write:  “We will not dwell on the importance of speech and music in the presence of premature infants because this must be dealt with extensively in other chapters, but…”

Is this manuscript intended to be a book chapter?

We deleted this sentence. We added a paragraph about music with the references indicated: intervention are especially difficult to fully describe due to the complexity of music stimuli (e.g., rhythm, pitch, tempo, harmonic, structure, timbre, jitter, shimmer, etc), variety of music experiences and factors due to music interventions. It is therefore difficult to know the effects of the rhythm itself. We only know that there is no music without rhythm, whereas there can be music without melody [35]. Music has often been effectively used in neonatal intensive care units, especially with high-risk infants [14]. Moreover, music is thought to improve neurodevelopment in premature infants by promoting synaptic plasticity and differentiation, activation, readjustment and growth of neurons [70]. A review of music therapy in NICU between 1970 and 2010 reveals previously unsuspected perceptual, adaptative and active engagement capacities of the preterm child during music therapy [71] The authors focus on interventions on music or auditory stimulation that incorporates musical elements such as rhythm or sounds based on the acoustic rhythmic intrauterine environment loke recorded womb sounds, the recorded mother’s voice, breathing sounds and heartbeats. The review shows that music has positive effects on the premature infant. He calms down, relaxes, and his stress level decrease. An another systematic review on Music based Intervention research published from 2010 to 2015 reports poor quality of music intervention studies [72]. The authors recommended to improve reporting quality, scientific rigor, and clinical relevance of music intervention research and used a seven-checklist component to advance scientific rigor and clinical relevance of music intervention research. 

And we also added a section Voice with references indicated by the reviewer: Linguistic research showed that lullabies of all cultures combine language information and use calming, rhythmic stimuli. Lullabies, with no tempo change, were used to reinforced non-nutritive sucking rate of premature infant. Contingent lullabies, such as pacifier-activated lullabies (PALs) increased pacifier sucking rate of premature infants [77], increased subsequent feeding rate [78] and shortened gavage feeding length when used at the specific gestation age of 34 weeks [79]. The rhythmic lullabies reinforced the sucking rate produced by the preterm infant. Consequently, the sucking rhythms is modified by lullabies: the more the preterm infant sucks the more the music will be provided. The premature infant can learn to suck-swallow-breath with music contingency.

In the section on Kangaroo Care, your manuscript content begins to extend far beyond rhythm and incorporates many other musical elements/stimuli.  This expansion makes it hard to determine which element or stimulus is eliciting the desired effect.  You also have not justified musical elements other than rhythm in the preceding sections.

 The kangaroo care is links to Intersensory redundancy. The firsts sentences of the section explain the multiplicity of rhythms in synchrony present in skin to skin contact.  We added the sentence: Mother–infant skin-to-skin contact, commonly known as kangaroo care underscore the importance of maternal body contact for infants’ physiological, emotional, and cognitive regulatory capacities [83].

Line 364 you write:  “Concerning the premature children in the group with kangaroo care and maternal singing, the authors observed that changes in Heart Rate Variability (HRV) indicated better autonomic stability and a calming effect: the decreased Low Frequency/ High Frequency ratio on the preterm infant with maternal singing was related to a calming effect.”

How did HRV change?  Please be specific.  What do you mean by “Low Frequency/ High Frequency ratio” and is this an effect observed on the infants, or in the infants?

 We have deleted the sentence

When describing Rhythm, Breath and Lullaby, you state:

“The music during intervention is typically improvised and based on elements such as slow tempo and repetition.”

What, specifically, is improvised?  Does this involve improvised singing?  Is this different than infant-directed singing?  Is the mother singing, or is the music therapist singing?

Or, does the improvisation involve improvised movement?  Or improvised playing of instruments?

You discuss the outcomes of this approach for parents – what about the infant?

And is this technique best applied to full-term or premature or medically-fragile infants?  Please clarify.

 We have simplified the sentence to highlight the rhythmicity of the stimulation and the close side of in utero stimulation

Carefully consider what you are recommending (again, to avoid mis-use).  Is a lullaby the only kind of song to sing to an infant?  The gist of infant-directed singing is to sing in a way that either reflects or beneficially modifies infant state.  Given this definition, is a lullaby always appropriate, or might other song styles also be appropriate?

 The purpose of this review is not to prescribe clinical recommendations. Lullabies have all the characteristics that are best perceived by the very young infant, preterm and even the fetus

When describing music therapy for preterm infants, your writing could be strengthened by reference to some of the leading international researchers in the field, notably:

Helen Shoemark

Deanna Hanson-Abromeit

Friederike Barbara Haslbeck

Lori Gooding

 Thank you for these references. We have considered the 4 references in the review.

  1. Empirical pilot study

-What do you mean by “this observational study?”  Does your study have a design?

We have modified: it is more of an empirical pilot study thanks you for the term: there are only ten dyads (mother and premature children) have been observed by video

-While a protocol diagram is always helpful, Figure 1 did not display clearly for me.

 We have slightly modified this figure

-When describing the mothers’ behavioural indicators, you mention “the mother’s non-bearing and active hand..” What is a non-bearing hand?   The mother does not touch her baby but may make other gestures (like taking their mobile phone, touching her chair…)

Were mothers given any definition of a lullaby?  Were mothers provided with any specific instructions on how to sing? The mothers are instructed to sing a familiar lullaby for 5 minutes with the instruction to sing continuously without stopping: they chose the familiar lullaby so as to repeat it if necessary; none of the 5 mothers sang the same lullaby. 

What is meant by “tonic-emotional manifestation?”  Ideally, this term should be clearly defined in a prior section of the manuscript.

For clarity we have removed this specific term for psycho-motor therapist

Regarding data collection, you mention:

“During the video analysis, for both phases (initial and test) and for each appearance of a given behavioural item, a start and end time marker was noted in order to obtain the time interval, i.e. the duration of each item (unit duration).”What was the time interval?  Did you use time sampling?  Did you measure duration or frequency of behavior?  Or percentage of time engaged in a particular behavior?  Please clarify. 

The duration of each behavior was measured to the tenth of a second

We calculated 3 dependent variables

- the unit duration of each item

- the total cumulative duration of these unit durations

- the number of times each item appears

Then we could calculate the average duration.

We have reworded all the paragraph

“We measured (in seconds) the time spent (via video analysis) (which the mother or the premature child for each item measured); we have called unit duration; we have added up all the different unit duration in order to have the cumulative duration in seconds.

In addition, we divided this total duration by the number of occurrences of the item (No percentage was needed because we had exactly 300 seconds pour each dyads)”

Line 434 What is desaturation?

We have specified « oxygen » desaturation

Figures 2 and 3 are helpful.  Do the asterisks indicate significant differences?  Please explain. Of course, asterisks indicate significant differences as indicated in the text; we added in the legend

You appear to have used an ANOVA (F test) to analyze the data.  How is the F test appropriate with such a small sample size and just two groups? Is the “significance” you obtained reliable in such a small sample?   This point should be addressed as a limitation.

A parametric test such as the Mann Whitney U Test seems more appropriate to study differences with such small numbers; we have therefore taken over the entire statistical analysis. We have inserted the error bars

Line 470, you write:  “Concerning the disorganised looks at the baby, the unit duration of each disorganised look at the baby lasts less in the group with a lullaby. Do you mean to say “disorganized looks of the baby..?”  You are talking about infant gaze here, not the mother’s gaze toward the infant, correct? Yes sorry, this is a translation error; we have corrected: It is the premature child’s disorganised gaze (or disorganized looks of the premature child).

You seem to have collected data on three infant gaze types, but only reported results for two gaze types (disorganized and eyes closed).  Results should be presented for all gaze types.

On video, it is difficult to know is the premature child is looking towards the mother (the visual system is not mature); we therefore considered 2 items relating to the gaze of the premature child 

Please provide a more in-depth description of the data for infant facial expression, even if just neutral.

In the part 2.2.3. Data analysis: the smile is described as upward movement of the lips with squinting of the eyes), the grimace as random and fluctuating contractions of the facial features and neutral face without any movement).

What was the importance or role of rhythm in the lullaby singing within your observational study?  You seem to have strayed significantly from your original point of the impact of rhythmic input on fetal and newborn well-being, as emphasized through much of the first 7 pages of the manuscript.  I don’t see yet that the review of literature is strongly related to the clinical case study.

 Skin-to-skin contact with lullaby is the only situation that offers to the premature babies in the NICU a synchronicity of stimulation that is identical to that present in utero. Inter-sensory redundancy is therefore present in this situation.

The conclusion needs revision to better broaden the reader’s understanding of the topic and raise questions/ideas for future research or clinical applications.  Right now, the conclusion appears to be a brief summary of the manuscript. 

 We have added the conclusion at the end of the pilot study, and we have reviewed the final comments

Some final thoughts to apply throughout the manuscript:

  • Throughout, when reviewing research findings, please clarify whether significant or non-significant (statistically or practically).
  • I see use of the words fetus, infant, toddler, and child. To avoid confusion (and again, misuse of recommendations) I recommend defining the age range(s) of interest and being consistent with terminology.
  • Many small type-o’s throughout the manuscript would be easily corrected through careful proofing or use of a good copy editor.
  • Strive for correct sentence structure and consistent verb tense throughout the manuscript.

 English is not our native language. We have made the corrections as quickly as possible, but we can have the text proofread by an English speaker if necessary

Reviewer 2 Report

#Comments to the author This manuscript elegantly reviewed studies on fetus' rhythm perception and production, medical interventions for preterm infants in NICU. Authors also introduced their exploratory study in preterm infants, and its findings were interesting to add on this topic. I believe this review paper will interest many readers from various fields, such as pediatrics, cognitive science, developmental science and neuroscience. Finally, the authors extended their previous review work (i.e., Provasi et al., 2014).  I have the following questions and comments especially on [Section 2: Our exploratory study], which I think the authors can address.    #Major comments on [Section 2: Our exploratory study] -Interesting study and findings. However, for better readability, I recommend making subsections for this part too (e.g., 2.1. Introduction, 2.2. Method, 2.2.1. Participants, 2.2.2. Procedure, 2.2.3. Data analysis, 2.3. Results, 2.4. Summary). -Throughout this section,  "skin-to-skin" contact is used instead of "Kangaroo Care(KC)". I wonder whether the current skin-to-skin contact intervention was different from KC so that the term was refrained from being used.   -Figure 1: Was the experiment conducted only once? If so, experimental flow on the experiment day might be informative for readers. Random assignment, contact & collection of informed consent can be described in the context instead of presenting in Figure 1. Abbreviation "SDF" in T0 was not clear. - Definition of cumulative duration and unit duration was difficult to understand. This relates to the following question.  - Y-axis for Figure 2 to 4 represents cumulative durations of the target behavior (initial phase + test phase, if my understanding is right). I wonder why the cumulative durations were chosen to examine an effect of lullaby.  I think comparing the behavioral changes between initial and test phase for each group is more straightforward to examine the lullaby effect on mothers' and infants' behavior.  -Figure 2: If 1) summation of gestures w/ contact and w/o contact is a total occurrence of mothers' gestures and 2) this study hypothesizes mothers in the lullaby group may increase contact gestures to infants, then analysis for "gestures with contact" is sufficient.   -Figure 3: Same comment as above. If 1) summation of looking at baby and looking away is a total measurement of mothers' looking behavior and 2) this study hypothesizes mothers in the lullaby group may look at their infants more, then analysis for "looking at baby" is sufficient.   -Figure 4: Same comment as above. If 1) summation of disorganised gaze and closed eyes is a total measurement of infants' gaze pattern and 2) this study hypothesizes the lullaby may promote infants' relaxation, then analysis for "closed eyes" is sufficient.   -Insertion of error bars are recommended for each Figure. -Line 470-472: It was confusing to follow. If this sentence describes "infants' disorganized gaze" during the intervention, then revision is necessary to avoid any confusion.     #Minor comments -I found some typos in the current manuscript. For example, line 25, line 102-103, line 373 etc. Thus, spelling check, English proofreading and format editing for reference lists are recommended during the revision. -The following work can be relevant literature in the subsection of Kangaroo Care: Feldman, R., Weller, A., Sirota, L., & Eidelman, A. I. (2002). Skin-to-Skin contact (Kangaroo care) promotes self-regulation in premature infants: sleep-wake cyclicity, arousal modulation, and sustained exploration. Developmental psychology38(2), 194.

Author Response

We thank reviewer 2 for these very pertinent remarks which will help us to significantly improve the paper. Please find our responses to your different point below, in green. We modified the manuscript accordingly.

Reviewer 3 Report

The current manuscript offers a review of rhythmic stimulation across modalities typically experienced in utero. The contributions of maternal heart rate, respiration, vocalisations, and footfalls on typical development are considered. The authors then describe the lack of these experiences for the premature infant being cared for in NICU, and some interventions that have used rhythmic stimulation in different modalities to improve infant physical and cognitive outcomes. In the second section of the manuscript, the authors describe some empirical pilot work looking at the impact of maternal singing to infants in NICU on infant behaviour. Whilst the review section of the paper is comprehensive, in its current format the pilot study presented seems somewhat disjointed from the review, and I have some issues with the presentation of early results. I lay out my concerns below, and hope that they are useful to the authors.

Major Comments:

  1. The rationale for the pilot study is not clear, and does not seem to fully follow from the literature review presented. Whilst the review focussed concretely on the role of rhythm, the rhythmicity of the intervention (maternal singing) is not obvious. It would be useful if the authors laid out clear aims and hypothesis. For example, is it the vocal rhythm of the lullaby that is hypothesised to change infant behaviour, or are the authors counting on these infants also experiencing more vestibular/proprioceptive cues from the (presumably at least marginally) moving mother? Is the tempo/rate/variability of the lullaby important? Explicitly defining the putative mechanisms of intervention would add clarity.
  2. The behaviour of the mothers during the test phase is not adequately described. In order to ascertain the impact of maternal rhythmicity on infant behaviour, it is critical to know what the mothers were doing during the test period. Presumably, the mothers did not sing constantly for five minutes in the lullaby condition, and it is not clear whether they were also moving in time with their singing. Whilst the tempo and variability of rhythm are highlighted in the review as critical to success of prior interventions, these features of maternal singing are not quantified here. Further, it is unclear as to the behaviour of the no-lullaby group – did these mothers still speak to their infant, perhaps using infant directed speech? It would be helpful to use the existing videos to quantify the rhythmicity of each mother, and the relationship between this metric and her infant’s behaviour.
  3. Given the emphasis of the literature review, it seems likely that the authors hypothesised an affect of maternal singing on desaturations and bradycardias, but these were not found. If this is the case, it would be useful if this hypothesis could be clearly stated, and the failure to find an effect of intervention on these outcomes elaborated upon. Bayesian statistics (e.g. readily done with default priors using JASP - https://jasp-stats.org/) could help to unpack if the null finding was due to low statistical power, or there is evidence for the null. It would be beneficial if the authors could unpack why they think their intervention did not improve this metric, where other rhythmic interventions have been successful.
  4. The reporting of the statistics is not clear. It was not easy to discern if the cumulative durations presented were the average of each group of infants, or the total for all five infants in each group added together. If the former, it would be very useful for them to be reported as means/medians, and for error bars to be added to the plots. If the latter, it would be very helpful to the reader for the total contributed by each mother to also be reported (e.g. in a table), perhaps as a series of case studies. Further, the statistical tests used to obtain the F values reported should be fully described.
  5. I very much enjoyed the breadth of modalities discussed in the review, but found it odd considering one of the key dependent variables of the study was ‘gestures with contact’, that there was not a section on rhythmic touch. Whilst stroking/caressing may not be possible in utero, it seems possible that this may be a common rhythmic experience for NICU infants. Given the finding that the lullaby condition elicited more touch, it would be useful to break down these results by rhythmic and arrhythmic touch. This would also speak to intersensory redundancy promoted as important by the authors in the review.

Minor Comments:

  1. The first paragraph of the introduction emphasises infant sensorimotor synchronisation, but the bulk of the literature discussed in the review focuses on stimulation that occurs to the infant (i.e. is not initiated by the infant). As it does not feel as intuitive for passive rhythmic stimulation (e.g. perception of maternal heart rate) to necessarily be social, the link, here and in section 1.2, with social interaction would benefit from being unpacked further.
  2. There are references to this manuscript being a chapter, but it is not clear as to what text the manuscript will be embedded in. Given the huge emphasis on voice and music in the intervention presented in section 2, it feels very odd that voice and music is consciously skipped over in section 1.3.4. This is also part of why the rationale for the study is not clear. Expansion of this section could help align the review and the pilot data.
  3. In Figure 1, ten minutes are shown before skin-to-skin started, but it reads as only five minutes in the text.
  4. Could typical ‘gestures without contact’ be described?
  5. Figures 2-4 all use different shading for the two groups, consistent coding would be helpful.
  6. Both looking at baby and looking away are analysed, but surely they are the inverse of each other. If so, it seems redundant to analyse both.
  7. On lines 470-471 it states ‘looks at the baby’ (i.e. insinuating the looks of the mother towards her infant), but it seems more likely this is meant to read ‘looks of the baby’ (i.e. from the infant towards the mother)?
  8. Figure 4 is captioned to show gaze towards the mother, but these data are not shown in the plot.
  9. The concluding paragraph seems to be more of a conclusion of the review section only, and does not integrate the findings of the new data presented.

Author Response

We thank reviewer 3 for these very pertinent remarks which will help us to significantly improve the paper. Please find our responses to your different point below, in green. We modified the manuscript accordingly.

Round 2

Reviewer 2 Report

I would like to thank the authors for their careful attention to the reviewer comments. I feel that they have been addressed clearly, and the edits to the manuscript improve clarity and impact.

My only remaining concern is results represented in Figure 2. It was evident that the total duration of gestures w/ contact and w/o contact is 5 min in both groups because p-values from the statistical analysis were the same (p = 0.016). This means, one of them is redundant data. If this study aims at examining positive effects of lullaby (and if lullaby groups increased contact gestures from the initial phase), then "gestures w/ contact" might be sufficient data to compare the two groups.

Minor comments:

1. I found some non-English articles in References. Adequate formatting seems necessary.  
2. Line 244: A sentence is starting with a reference number.   

Author Response

Thank you for all the very constructive comments that helped to improve the paper.
